# The Quest for Child-Friendly Carrier Materials Used in the 3D Semi-Solid Extrusion Printing of Medicines

**DOI:** 10.3390/pharmaceutics15010028

**Published:** 2022-12-21

**Authors:** Eveline E. M. van Kampen, Sejad Ayyoubi, Luc Willemsteijn, Kjeld J. C. van Bommel, Elisabeth J. Ruijgrok

**Affiliations:** 1Erasmus MC, University Medical Center Rotterdam, Department of Hospital Pharmacy, 3015 GD Rotterdam, The Netherlands; 2Erasmus MC-Sophia Children’s Hospital, University Medical Center Rotterdam, Department of Hospital Pharmacy, 3015 CN Rotterdam, The Netherlands; 3The Netherlands Organization for Applied Scientific Research (TNO), 5656 NL Eindhoven, The Netherlands

**Keywords:** semi-solid extrusion, three-dimensional printing, children, dual asymmetric centrifugation, pharmaceutical preparations, pharmaceutical technology

## Abstract

This work gives a brief overview of carrier materials currently used in pharmaceutical studies on the three-dimensional (3D) semi-solid extrusion (SSE) printing of medicines for pediatrics. The suitability of using these carrier materials in pediatric formulations, concerning safety and toxicity, was reviewed by consulting the ‘Safety & Toxicity of Excipients for Pediatrics’ (STEP) database and the Food and Drug Administration (FDA) regulations. In the second part of this work, carrier materials were tested on their ability to form a semi-solid mixture with lactose by dual asymmetric centrifugation (DAC) and printing by SSE. With the combination of theoretical and experimental studies, this work will guide research toward grounded decision-making when it comes to carrier material selection for pharmaceutical pediatric 3D SSE printing formulations.

## 1. Introduction

Administering the correct dose of medication to children is a complex task. Children, compared to adults, need lower doses and a flexible dosage form as they experience physical growth and their organ functions mature [1]. Consequently, many dose corrections are performed through splitting, crushing and dissolving tablets or administering intravenous liquids orally. Cross-contamination, inaccurate dosing and altered absorption are associated risks with these manipulations [2]. In the quest for better medication for children, 3D printing can provide a good option. With 3D printing, dose-flexible and age-appropriate solid oral dosage forms can be manufactured [3]. By pharmaceutical printing, the active pharmaceutical ingredient (API) is added to a mixture of the carrier material, forming the base of the formulation and excipients, enhancing properties such as stand-up or dissolution profiles. After mixing all compounds, an object created by computer-aided design (CAD) software is printed layer-by-layer [4]. The field of printing child-friendly, dose-flexible solid oral dosage forms is evolving as multiple 3D printing techniques are studied [3].

Extrusion-based 3D printing is a frequently used method in the 3D printing of drugs. With fused deposition modeling (FDM), API-loaded filaments are pushed through a heated nozzle, where they are melted and extruded. Examples of FDM-printed child-friendly drugs are chewable candy-like formulations with indomethacin [5], minitablets with baclofen [6] and nifedipine [7]. With semi-solid extrusion (SSE), a syringe is filled with a molten or semi-solid formulation, after which the material is pressed through a nozzle by a piston or back pressure. SSE operates at lower temperatures than FDM and therefore is suitable for thermally sensitive APIs. Proof-of-concept studies showed successful printing of different chocolate-based formulations with ibuprofen or paracetamol for pediatric use [8]. Other studies report the production of 3D-printed gummies with ranitidine and lamotrigine, low-dose furosemide, and sildenafil tablets for children [9]. As sildenafil was tested in a clinical setting, this gives insight into the excellent feasibility of using SSE printing in a hospital pharmacy as small-scale manufacturing.

For the development of a pediatric formulation, carrier material and excipients should be considered carefully. When it comes to patient safety, and more specifically pediatric patient safety, the materials should be well tolerated by children [10]. As their physiological systems are still developing, certain compounds (e.g., propylene glycol or ethanol) cannot be metabolized by children as they can by adults [11]. So even compounds generally regarded as safe (GRAS) by the FDA might not be tolerated. The STEP database, set up by the European Pediatric Formulation Initiative (EuPFI), covers excipient safety data on pediatric use from peer-reviewed scientific journals, government reports and databases. Besides patient safety, material properties are critical for developing a pediatric formulation. The most relevant material property in 3D SSE printing is the melting temperature of the carrier material, as the processing temperature should remain below the degradation temperature of the API. Degradation of API could also occur when the API is exposed for a longer period to elevated temperatures even when they remain below the degradation temperature.

The objective of this work is to give an overview of carrier materials used in SSE 3D printing research for pediatric use and to evaluate their theoretical suitability and experimental extrudability. We describe the selection of a suitable carrier material(s) to print child-friendly mini-tablets using the SSE 3D print method. First, suitable carriers were identified by a literature search, then their safe application in children was assessed theoretically, and finally, their extrudability was determined in practical experiments.

## 2. Materials and Methods

### 2.1. Carrier Material Selection from Literature

Suitable carrier materials were selected by performing a Pubmed literature search (Appendix A) and screening the ‘Handbook of Pharmaceutical Excipients’ [12]. The literature search focused on materials used for pharmaceutical SSE 3D printing of medication for pediatrics. Carrier materials were selected if reported as 3D printed or extruded for pediatric purposes and had a melting temperature or glass-transition temperature between 30 °C and 80 °C.

### 2.2. Assessment of Suitability for Pediatric Use

The selected carrier materials were assessed on their suitability for pediatric use by screening the STEP and US FDA Inactive Ingredients databases for each carrier material [13,14]. Materials were included as appropriate if the FDA currently approved their oral use. The STEP database was also consulted, as the FDA approval for oral use does not explicitly specify safe for pediatric use. The STEP database includes an up-to-date overview of the literature on the safety and toxicity of carrier materials used in children. Available data on the acceptable daily intake (ADI) in children described in this database was noted in the results.

### 2.3. Extrudability Screening

From the 27 carrier materials found, we assessed the extrudability of ten available carrier materials by one preparation and extrusion process for equal comparison. For this selection, the available information on pediatric safety or FDA approval for in-human-use was not used to create specific inclusion and exclusion criteria. Gelling agents were excluded from the experiments as preparing tablets based on gelling agents would result in an additional drying step unincorporated in our process.

Ten materials were evaluated using an experimental setup of the dual axial centrifuge (DAC) and SSE 3D printing. DAC is preferred over mixing by mortar and pestle to increase the reproducibility of the process. First, the materials were mixed and melted by DAC (Section 2.3.2), second, the syringe was filled (Section 2.3.3), and third, the mixture was extruded by our SSE 3D printer (Section 2.3.4). Each step is described in detail below. There was an evaluation step if the process was successful or unsuccessful, visualized in a decision tree in Figure 1.

#### 2.3.1. Materials

Methacrylate-Copolymers (Eudragit E PO) were obtained by Evonik. Poloxamer 407 (Kolliphor^®^ P407) was kindly donated by BASF (Ludwigshafen, Germany). Polyethylene glycol 4000 (PEG 4000) (EMPROVE^®^ ESSENTIAL Ph Eur) and Polyethylene oxide (PEO) were purchased from Sigma-Aldrich (Zwijndrecht, The Netherlands). Adeps solidus (Witepsol^®^ H15) was obtained from BÜFA. White beeswax (Cera Alba) was purchased from Fagron (Capelle aan den Ijssel, The Netherlands). Glycerol dibehenate (Compritol^®^ 888 ATO), glycerol distearate (type I) (Precirol^®^ ATO 5), glycerol monostearate 40–55 (Type I) (Geleol™ mono and diglyceridedes NF) and stearoyl polyoxyl-32 glycerides (Gelucire^®^ 50/13) were kindly donated by Gattefossé (Saint-Priest, France). Lactose (SorboLac^®^ 400) was kindly donated by MEGGLE Group (Wasserburg am Inn, Germany).

#### 2.3.2. Mixing and Melting

The carrier material and lactose (40:60 *w*/*w*) were weighed to a total of 55 g and placed in a disposable plastic Speedmixer™ cup (185 mL). Subsequently, the cup was placed in a SpeedMixer™ (DAC 150.1 FVZ-K) and mixed by DAC at 3500 rpm for 1 min. Afterward, the temperature of the semi-solid mixture was measured. In DAC, the Speedmixer™ cup rotates around two axes causing a particle flow that generates friction between the individual particles. This friction generates enough heat to melt the carrying material. Adding an excipient with a high melting temperature (Tm), in this case, lactose (Tm = 140 °C) enhances the particle friction during DAC, resulting in heat development. The DAC step was repeated until a visually homogenous mixture with a constant temperature was obtained.

The mixture was transferred to the pre-heated stainless-steel syringe of the SSE 3D printer once it visually appeared homogeneous, and the temperature was constant above the melting point of the carrier material. If the mixture was not molten or the temperature was not above the melting point, the mixing and melting in the DAC continued for an extra minute at 3500 rpm. If this repetition was unsuccessful, the percentage of lactose was increased stepwise. The adjusted formulation was mixed until the desired consistency and temperature, resulting in successful mixing. If neither the prolonged DAC mixing nor the increased amount of lactose led to an enhanced texture, mixing was declared unsuccessful.

#### 2.3.3. Syringe Filling

Heating the stainless-steel syringe to the melting temperature of the carrier material before filling prevented the material from hardening while filling the syringe. The pre-heated syringe was placed on a plastic holder. This holder was the size of the Speedmixer™ cup with an opening in the middle, matching the diameter of the syringe. The holder with the syringe was placed on top of the molten formulation in the Speedmixer™ cup. By pressing the holder downwards with a SpeedPress™, the molten formulation was pushed upwards filling the syringe [15].

#### 2.3.4. Melt-Extrusion 3D Printing

Figure 2 illustrates the setup of the SSE 3D printer used to determine the extrusion of the mixture. First, the syringe (A) loaded with the molten formulation was placed in the syringe heater (B). The heater was adjusted to the melting temperature of the carrier material. Second, a metal nozzle (C) with a diameter of 0.4 mm was attached to the syringe. The nozzle was heated by a separate heating block (D) to the melting temperature of the carrier material, to prevent solidification of the material in the nozzle. Third, a mechanically controlled piston (E) was attached. With the piston, the material was extruded through the nozzle and the applied force was logged. The maximum force of this printer setup was 400 N. Finally, the printer head, controlled by a computer (F), moved in three dimensions (*x*-, *y*-, and *z*-axis). If the printed formulation did not adhere to the printing platform (G), as the formulation cools too quickly, the platform could be heated with a separate heating bed (H). On the contrary, if the formulations did not solidify due to residence heat, air cooling (I) could be applied to enhance the solidification of the tablet.

In the experimental part of this study, the formulations were extruded onto a glass plate or a piece of baking paper. Instead of printing a specific shape, the software Pronterface monitored and sent commands to the 3D printer. In this experimental stage, the command to extrude was given manually and not by running any G-code.

## 3. Results

### 3.1. Carrier Material Selection from Literature

From our literature search, 27 carrier materials were found that complied with the selection criteria of being 3D printed or extruded with a melting or glass temperature in the range of 30–80 °C (Table 1). All 27 materials were printable below a temperature of 80 °C using SSE. To structure the different materials, we organized them into four categories: polymers (e.g., polyethylene glycol), fats (e.g., beeswax), gelling agents (e.g., hydroxypropyl methylcellulose), and others.

The carrier materials show a wide range of melting and glass transition temperatures, ranging from 30 °C (Tm of adeps solidus) to above 80 °C (glass transition temperature (Tg) of polyvinylpyrrolidone K12 (PVP)). Formulations based on different carrier materials not only differed in processing temperature but also in release profile. Use of highly water-soluble PEG resulted in tablets with a fast and complete release profile. Tablets made from highly lipophilic and water-insoluble fats tended to show a more sustained release, depending on the API. Most of the 27 carrier materials were printed with different types of API at different concentrations, ranging from nearly 1% (*w*/*w*) of ibuprofen and acetaminophen [8] to 50% (*w*/*w*) of praziquantel [16].

There was also a variety of preparation methods of the semi-solid mixture. To compare the carrier materials equally, we selected ten available carrier materials for the experimental part of this study.

### 3.2. Assessment of Suitability for Pediatric Use

In general, it can be stated that there is a limited amount of information on the safety of carrier materials for pediatric use. The expected concentration of carrier material is 30–70% in 3D-printed tablets weighing 50–250 mg. This amount should remain below the toxicity limits for children. From the selected of 27 carrier materials, 18 are approved by the FDA to use in food or drugs. Six carrier materials are present in the STEP database; poloxamer 407, polyethylene glycol (PEG) 4000, polyvinylpyrrolidone (PVP), carboxymethylcellulose (CMC), hydroxypropyl methylcellulose (HPMC), and glycerol dibehenate. For PEG 4000, HPMC and PVP, the ADI is indicated. Of the selected 27 carrier materials, only three are both present in the STEP database and FDA-approved: poloxamer 407, PEG 4000 and HPMC.

PEG is a commonly used excipient in pharmaceutical products. Small amounts of PEG are added to liquid formulations to adjust the viscosity of liquids or alter the solubility of APIs. Relatively large amounts of PEG are used in solid dosage forms such as tablets and suppositories to create a solid dispersion. Due to its laxative properties, PEG has a maximum ADI of 10 mg/kg body weight when used as an excipient for children. High doses of PEG 4000, up to 0.8 g/kg body weight, are used in children of one month and older when these laxative properties are intended.

PVP is the second carrier material for which an ADI is described. The World Health Organization (WHO) defined an ADI of 0–50 mg/kg body weight for PVP [13]. It is primarily used as a binder in oral dosage forms. Although there is information on ADI, there is limited information available regarding clinical data in the pediatric population [13].

Another suitable material is poloxamer 407. Although clinical data on oral administration and ADI is lacking, there is data focused on the parenteral administration in children with sickle-cell disease [17,18]. However, the absorption in the gastrointestinal tract should be studied in further detail before general use. 

### 3.3. Extrudability Screening

Ten available carrier materials, from the 27 carrier materials found in the literature, were assessed on their extrudability with our printing process. Gelling agents were excluded from the experiments as their preparation method needed an additional drying step unincorporated in our process. Following the decision tree, it was determined for each of the 10 materials if a process step could be executed successfully or not. Table 2 shows the outcome of this decision tree, where a green √ indicates a successful result, an orange ~ indicates it was partly possible and a red × indicates an unsuccessful outcome. The following sections describe the principal findings for each process step.

#### 3.3.1. Mixing and Melting

All formulations containing a carrier material plus lactose reached a temperature above the melting temperature or glass transition temperature of the carrier material after DAC. Nearly all formulations (PEG 4000, poloxamer 407, beeswax, adeps solidus, glycerol dibehenate, glycerol distearate, glycerol monostearate and stearoyl polyoxyl-32 glycerides) formed a semi-solid mixture after DAC. Exceptions were the formulation with PEO 100,000 remaining too viscous to fill the syringe and Eudragit^®^ E PO resulting in hard round lumps which created cracks in the Speedmixer™ cup after DAC. As these formulations containing PEO and Eudragit^®^ E PO did not form an adequate semi-solid mixture, they were considered unsuitable for SSE in combination with DAC.

In general, we observed that fat-based formulations needed a higher percentage (*w*/*w*) of lactose (55–70%) compared to polymer-based formulations (40–55%) to reach a temperature above the melting or glass transition temperature.

**Table 1 pharmaceutics-15-00028-t001:** Carrier material selection from the literature.

Carrier Material	Melting or Glass Transition Temperature (°C)	Pharmaceutical Considerations	Pediatric Toxicology—STEP-Database—FDA Approved	Relevant Extrusion and/or 3D Print Example(s) in Literature
**Polymers**				
Methacrylate-Copolymers (Eudragit)	54 (Tg) (Eudragit^®^ EPO) 63 (Tg) (Eudragit RL^®^) 64 (Tg) (Eudragit RS^®^)	Different release profiles (sustained release, enteric), based on grade. Wide applicability in the pharmaceutical industry	Not included in the STEP database [13] FDA approved for oral use [14]	Tablets Eudragit RS with 40% theophylline were printed using direct extrusion at a printing temperature of 80 °C [19] Eudragit EPO: TEC: Theophylline (50: 3.25: 46.75) was extruded at 90 °C and printed using FDM at 145 °C [20]
Poloxamer 407	52–57 (Tm) [12]	Used in soft capsules, tablets, and suspensions [21] Complete drug release within 2 h [22]	FDA-approved for oral use [14] GRAS [13] Included as an excipient for pediatric formulations by the WHO [13] No data on acceptable daily intake [13]	Tablets containing 5% (*w*/*w*) pantoprazole made with HME-FDM printed at 60 °C [22]
Polyethylene glycol (PEG) Mn 4000/6000	50–58 (Tm) [12]	Complete and fast-release profile [23]	ADI: 10 mg/kg body weight [13] FDA-approved for oral use [14] Used as laxative agent up to 20 g/day (Children aged 8–18 years) [24]	Puerarin: PEG4000 (1:3–6) tablets made using melt extrusion 3D printing at 54–57 °C [23] Tablets containing 10% (*w*/*w*) pantoprazole with PEG6000 were printed at 60 °C with HME-FDM [22,25]
Polyethylene-co-vinyl acetate	43 (Tm) (40% VA) 73 (Tm) (28% VA)	Slow and incomplete release profile [26] Non-degradable [27]	Not included in the STEP database [13] Not FDA-approved for oral use [14]	Extrudes with 10–50% (*w*/*w*) metoprolol produced via hot-melt extrusion at 60 °C (VA content 28% and 40%) [26] 50% (*w*/*w*) metoprolol extrudes with EVA (VA content 9–40%) were extruded with HME between 70–110 °C [28]
Poly(lactic-co-glycolic acid)	30–60 (Tg) [29]	Widely used in pharma applications Biocompatible/biodegradable	Not included in the STEP database [13] FDA-approved for oral use [14]	Drug delivery patch with PLGA 3D printed at 140 °C [30].
Polycaprolactone	50–60 (Tm) [31]	Slow and incomplete release profile [32] Biodegradable [31] Water insoluble [31]	Not included in the STEP database [13] Not FDA-approved for oral use [14]	Tablets containing 10–20% (*w*/*w*) ciprofloxacin produced with HME-FDM at 130–170 °C [32] Coating with polycaprolactone printed at 58 °C [33]
Polyethylene oxide (PEO) 100,000	65 (Tm) [34]	Non-ionic, hydrophilic and uncross-linked polymer [35]	Not included in the STEP database [13] FDA-approved for oral use [14]	Syringe-based 3D printing and piston-based extrusion. Drug/PEO ratio (50/50 and 80/20) at 60 °C and 80 °C [36]
Polylactic acid	50–80 (Tg) [37]	Biodegradable, thermoplastic polymer [38]	Not included in the STEP database [13] GRAS FDA-approved for oral use (Lactic acid) [14]	Only printed with relatively high temperatures so far (e.g., ≥135 °C) [39]
Polyvinyl caprolactam-polyvinyl acetate-polyethylene glycol graft copolymer (Soluplus^®^)	70 (Tg) [40]	Enhancement of bioavailability [40]	Exposure of 114 mg/kg is regarded as safe [40] Not FDA-approved for oral use [14] Not included in the STEP database [13]	Tablets containing dutasteride, Lutrol^®^ F68, and Soluplus^®^ (1:10:89) were printed at 160 °C using a hot-melt pneumatic dispenser [41]
Polyvinylpyrrolidone (PVP) K12	108 (Tg) [42]	Water-soluble Immediate release (100% within 15 min) [22]	ADI: 0 to 50 mg/kg body weight [13] Clinical trials in the pediatric population [13] Not FDA-approved for oral use [14]	Printed at 79–87 °C 10–30% pantoprazole and 10–25% TEC with HME-FDM [22]
**Fats**				
Adeps solidus (Witepsol^®^)	30–45 (Tm) [12]	Low melting point (storage conditions) Used in suppositories	Not FDA-approved for oral use [14] Not included in STEP database [13]	N/A
Beeswax, white	60–67 (Tm) [12]	Slow and incomplete release profile [43]	Not FDA-approved for oral use [14] Used as an excipient in tablets and capsules Not included in the STEP database [13]	Tablets containing 5% (*w*/*w*) fenofibrate with different infills made with a piezoelectric inkjet printer at 90 °C [43]
Glycerol dibehenate (Compritol^®^)	65–77 (Tm) [12]	Slow and incomplete release profile Dissolution profile alters over time [16]	FDA-approved for oral use [14] Only clinical data on topical use [13] No data on acceptable daily intake [13]	Extrudes containing 50% (*w*/*w*) praziquantel and 1% SiO_2_ produced with HME at a temperature just below the melting point [16]
Glycerol distearate (Precirol^®^ AT05)	50–60 (Tm) [44]	Slow and incomplete release profile	Not included in STEP database [13] Not FDA-approved for oral use [14]	Extrudes containing 15–20% (*w*/*w*) phenylpropanolamine made with HME at 60 °C [45]
Glycerol monostearate (Geleol™)	54– 64 (Tm) [46] 55–60 (Tm) [12]	Slow and incomplete release profile [16] Dissolution profile alters over time [16]	Not included in STEP database [13] FDA approved for oral use as an inactive ingredient [14]	Extrudes containing 50% (*w*/*w*) praziquantel produced with HME at a temperature just below the melting point [16]
Glycerol tripalmitate	66–67 (Tm) [16]	Slow and incomplete release profile [16] Dissolution profile alters over time due to unstable modifications [16]	Not included in STEP database [13] Not FDA-approved for oral use [14]	Extrudes containing 50% (*w*/*w*) praziquantel produced with HME at a temperature just below the melting point [16]
Paraffin (solid)	50–61 (Tm) [12]	Slow and incomplete release profile [16]	Not included in STEP database [13] FDA approved for oral use as an inactive ingredient [14]	Extrudes containing 50% (*w*/*w*) praziquantel produced with HME at a temperature just below the melting point [16]
**Gelling agents**				
Carboxymethylcellulose (CMC)	60–80 (solid-gel) [47]	Water-soluble, anionic cellulose-derivative [48] Biocompatible	FDA approved for oral use as an inactive ingredient [14] No ADI available [13] Widely used in pharmaceutical products	Tablets containing levetiracetam printed and processed at ambient temperature using semi-solid extrusion 3D printing [49]
Carrageenan	40–60 (solid-gel conversion) [50]	High molecular weight polysaccharide [51]	Not included in STEP database [13] Widely used in the food industry as a gelling, thickening, emulsifying and stabilizing agent. FDA-approved for oral use [14]	Printed at temperatures below 80°C with a liquid feed 3D printer [52]
Gelatin	22–26 (solid-gel conversion) [53]	Biocompatible polymer [54] Thermo-reversible gelation	Not included in STEP database [13] Widely used in food and pharma FDA-approved for oral use [14]	Printed at ambient temperature with a bioprinter [55]
Hydroxypropyl methylcellulose (Hypromellose, (HPMC))	55–77 (solid-gel) [56]	Safe biopolymer [57] Sustained release profile [58] Thermo-reversible gelation [57]	ADI: 0 to 25 mg/kg body weight as a food additive (sum of total modified celluloses) [13] GRAS [13] FDA approved for oral use as an inactive ingredient [13] Widely used in pharmaceutical products [57]	Semi-solid extrusion of tablets containing naftopidil after processing at 90 °C [25,59] Tablets with HPMC were made after processing at room temperature and cooling in the refrigerator at 4 °C [60] Semi-solid extrusion of tablets containing theophylline after processing at 80 °C [58] Gastro-floating tablets with dipyridamole prepared and printed at room temperature [61] Syringe loaded at 80 °C and printed with extrusion-based syringe printing at ambient temperature [62]
Methylcellulose (MC) A4M	60–80 (gelation)	Biocompatible Sustained release profile [63]	Widely used in food and pharma [64] Not included in STEP database [13] FDA approved for oral use as an inactive ingredient [14]	Semi-solid extrusion of tablets containing theophylline at ambient temperature after processing at 70 °C [63]
Pectin	15–50 (gelation) [65]	Biocompatible, gelling properties [66]	FDA approved for oral use as an inactive ingredient [14] Widely used in food and pharma Not included in STEP database [14]	Hydrogel printed at 25 °C after processing at 65 °C [67]
Starch (e.g., corn starch, waxy maize starch, potato starch)	75 (gelatinization) [68]	Safe biopolymer Drug release dependent on the starch type [69]	Widely used in pharmaceutical products [70] FDA approved for oral use as an inactive ingredient [14] Not included in STEP database [13]	Tablets containing ibuprofen printed using syringe-based 3D printing at ambient temperature after processing at 90 °C [69]
Xanthan gum	60 (HME processing temperature) [71]	Non-toxic, biodegradable polymer [72] Stable over a broad pH range	Not included in STEP database [13] Used in several commercial products FDA-approved for oral use [14]	Xanthan gum in combination with ethyl cellulose at an extrusion temperature of 60 °C [71]
**Other**				
Chocolate	29–30 (Tm)	Ethical and safety issues because of ‘candy-like formulation’ Good palatability Low melting point (storage conditions)	Food product High in sugars and saturated fat	Different attractive shapes with paracetamol and ibuprofen (different formulations) printed with an extrusion printer at 45 °C [8,73]
Lauroyl Polyoxyl-32 glycerides (Gelucire^®^ 44/14)	42.5–47.5 (Tm)	High Hydrophilic-Lipophilic Balance (HLB) Enhances the solubility of hydrophobic APIs [74] Tablets can be considered conventional-release tablets [9]	Not included in STEP database [13] Polyoxyl stearates are listed in the US FDA Inactive Ingredients database for several administration routes and dosage forms [14]	S-SMEDDS formulations (Gelucire^®^ 44/14, Gelucire^®^ 48/16, and Kolliphor^®^ P 188) loaded with fenofibrate or cinnarizine were printed at 65 °C [75] Suppositories with Gelucire^®^ 44/14, coconut oil, and tacrolimus were printed with SSE at 42 °C [76]
Polyoxyl-32 stearate (type I) NF (Gelucire^®^ 48/16)	46–50 (Tm)			Tablets with furosemide (2/10 mg) and sildenafil (4 mg) were printed using SSE at 44 °C [9] Suppositories with Gelucire^®^ 48/14, coconut oil, and tacrolimus were printed with SSE at 48 °C [76]
Stearoyl polyoxyl-32 glycerides (Gelucire^®^ 50/13)	46–51(Tm)			Tablets with ricobendazole were printed using SSE at 49 °C [74]

**Table 2 pharmaceutics-15-00028-t002:** Outcomes of extrudability screening per formulation of carrier material and lactose. (√ in green means successful, ~ in orange means suboptimal and a red × means unable to process).

Carrier Material	Lactose (% *w*/*w*)	Mixing	Filling	Extrusion	Measured Temperature after DAC (°C)	Minutes DAC	The Temperature of the Syringe and Nozzle (°C)
**Polymers**							
Methacrylate-Copolymers (Eudragit^®^ E PO)	40	×	n/a.	n/a.	40	1	n/a.
Poloxamer 407	42	√	√	√	>70	2	57
Polyethylene glycol (PEG) Mn 4000	55	√	√	√	67	4	75
Polyethylene oxide (PEO) 100,000	40	×	n/a.	n/a.	>100	2–3	n/a.
**Fats**							
Adeps solidus (Witepsol^®^)	60	√	√	√	44	2	35
Beeswax, white	70	√	~	√	80	3	63
Glycerol dibehenate (Compritol^®^)	70	√	√	√	70	1	70
Glycerol distearate (Precirol^®^ AT05)	55	√	√	√	58	2	57
Glycerol monostearate (Geleol™)	70	√	~	√	84	3	64
**Others**							
Gelucire^®^ 50/13	40	√	√	√	46	3	46

#### 3.3.2. Syringe Filling

Filling the syringe using the Speedpress™ is the following tested process step once a semi-solid mixture of carrier material and lactose was obtained after DAC. It was successful for PEG 4000, poloxamer 407, adeps solidus, glycerol dibehenate, glycerol distearate, and Stearoyl polyoxyl-32 glycerides. Fats with a relatively high melting point (e.g., glycerol monostearate) were challenging to process as they hardened quickly at room temperature. While being centrifuged with DAC, the material is pushed upwards along the sides of the Speedmixer™ cup. As the material solidified fast, a thin layer of solid material was formed, so the holder of the Speedpress™ rested on the remaining material. Accordingly, the holder could not be pushed down, and the syringe not filled. In contrast, the white beeswax formulation was so liquid that it leaked from all sides of the holder without filling the syringe.

#### 3.3.3. Melt-Extrusion 3D Printing

Successfully filled syringes were placed in the printer. To retain a semi-solid mixture, the stainless-steel syringe and nozzle head were both (pre-)heated to the melting or glass transition temperature of the carrier material. Experiments showed that each material that could be loaded into the syringe was suitable for extrusion using the 3D printing setup. Suitable extrusion was assessed by qualified operators and was defined by adequate flow of semi-solid material coming out of the printer below the force limit of 400 N.

## 4. Discussion

This study aimed to identify suitable carrier material(s) to print child-friendly tablets with SSE 3D printing. In the literature, we found 27 different carrier materials used in previous studies on the 3D printing of pediatric drugs. When formulating child-friendly oral medication, safety should be one of the main considerations. Of the 27 materials found, only six were described in the STEP database [13], of which the acceptable daily intake was reported for PEG 4000, PVP and HPMC. The lack of knowledge on the remaining carriers does not exclude their use in pediatric formulations, but currently their toxicity is not fully assessed. These knowledge gaps must be addressed further when considering these materials for future use [10]. The STEP database is a usable resource but far from complete.

Several questions on the use of carrier materials for the production of pediatric drugs remain. Information on the safety and toxicity aspects of carrier materials is limited, as stated by Quodbach et al. [77]. Even if the information is available, it might not cover its use for the 3D-printed medicines. In most cases, the safety evaluation is based on small amounts of material used as coatings and not in large amounts as carrier materials in 3D printing. Side effects become more important in the latter case. For example, using PEG up to 20 g/day is considered safe for children between 8 and 18 years. However, its laxative effects might be unfavorable in combination with specific APIs [24]. Besides the addressed constraints of the available literature, it becomes clear that the limited amount of information could hamper the development of safe 3D-printed pediatric formulations. This limitation emphasizes the need to extend a validated database, such as the STEP database, listing information on the safety and toxicity of excipients for pediatrics.

We created four categories to streamline the decision-making of child-friendly carrier materials; polymers, fats, gelling agents and others. The found studies used different techniques to prepare a semi-solid mixture, so we selected ten available carrier materials to evaluate in the experimental part of this study. We did not examine gelling agents in the experimental part as they need to be mixed with water, and some need to dry to solidify. In the case of ad hoc production in a hospital setting, using gelling agents in 3D-printed formulations might be too complicated as it is time-consuming. Some authors describe the use of chocolate as a carrier material. Out of ethical considerations, we do not consider this a realistic option used in daily clinical practice. The association of chocolate with candies might lead to an extensive intake of the final pharmaceutical product with the risk of overdosing children. Four polymers, five fats and one type of glyceride (Stearoyl polyoxyl-32 glycerides, Gelucire^®^) were evaluated by our SSE process.

The ten selected materials differed in their DAC processing requirements. For instance, fats needed more lactose to create enough particle friction to reach their melting temperature than polymers. The uniform shape of the saturated fatty acids might result in a tight packing and a crystalline-like structure, making it harder to transform the material into a semi-solid mixture. This transition can be enhanced with higher friction caused by increasing the amount of lactose. Most of the tested mixtures of carrier material and lactose could be processed by DAC in combination with SSE below 80 °C.

For two of the tested materials, it was impossible to create a semi-solid mixture by DAC either the mixture became too viscous (PEO) or too solid (Eudragit^®^). Picco et al. successfully mixed their formulation of PEO with olanzapine by using DAC, but they dissolved PEO in water to mix [36]. Eudragit^®^ was successfully printed by Kuźmińska et al., but they lowered the viscosity of their formulation by adding plasticizers such as triethyl citrate. Nevertheless, the process temperature remained above our set criteria of 80 °C [19]. Therefore, it seems that Eudragit^®^ is more suitable to process at higher temperatures [20]. From the four polymers tested in the experimental part, PEG 4000 and poloxamer 407 could be extruded by the SSE 3D printer. The success of printing PEG 4000 is consistent with the work of Li and others, even though the mixing and melting steps differ [23]. Instead of using DAC, they mixed in a pestle and mortar for 10 min and heated and stirred the mixture at 70 °C on a heating plate before SSE. Several studies report this method of mixing and melting by hand [22,23]. We prefer DAC to create a homogenous semi-solid mixture in a reproducible way. Poloxamer 407 is described in the production of thermo-sensitive drugs, shown by Kempin et al., which suggests its suitability in SSE.

Mixing by DAC and the extrudability was visually assessed by qualified operators. Although this is a qualitative assessment, this way of determining if the SSE mixture was molten, resulted in successful semi-solid mixtures for SSE. The temperature of the semi-solid mixture was measured quantitative. Furthermore, the force was measured during extrusion and did not exceed the limitations of the printer.

The majority of the selected fats could be processed successfully in the experimental phase. Exceptions were beeswax and glycerol monostearate, as filling the syringe was challenging with these formulations. White beeswax was too liquid, and glycerol monostearate solidified too quickly. Kyobula et al. used white beeswax to print different geometries to tune drug release and dissolution profiles when printing with a piezoelectric inkjet printer [43]. Their work suggested white beeswax as a suitable carrier material. On the contrary, the mixture of lactose and white beeswax did not solidify after SSE in our experiments and could not be clarified. A similar result was observed when extruding adeps solidus. By remaining soft at room temperature, the stability of the end product might be at risk. Adding other excipients to overcome this issue was not part of this study.

Here, we describe the first step in selecting child-friendly carrier materials for SSE printing. By adding more compounds to the mixture, e.g., API, the mixing and melting behavior of the powder mix in the Speedmixer™ might change or lead to different kinds of semi-solid mixtures, influencing the extrusion behavior. Formulating a final product containing API might lead to a different outcome than described here, as particle size and compatibility of the carrier material and API may influence the process. Each combination and addition of API should be further studied.

## 5. Conclusions

In this study, we aimed to provide a guide on the selection of carrier materials for manufacturing 3D-printed pediatric drugs. We have described 27 different carrier materials which might be suitable for SSE printing between 30 °C and 80 °C. We have assessed their available safety and toxicity data. The STEP database is useful but contains limited information. Furthermore, we have evaluated the extrudability of ten selected materials to assess consistency with the literature using DAC and our SSE 3D printing process. The results showed that studies should be critically reviewed, depending on the preparation method of the semi-solid mixture. Both PEG 4000 and poloxamer 407 performed well in the extrudability experiments, and their use is generally considered safe in children. Six other carrier materials were successfully printed. A toxicity assessment should be conducted before using them in pharmaceutical 3D printing for pediatric use. Although future studies are needed, this study supports implementing SSE printing of child-friendly drugs.

## Figures and Tables

**Figure 1 pharmaceutics-15-00028-f001:**
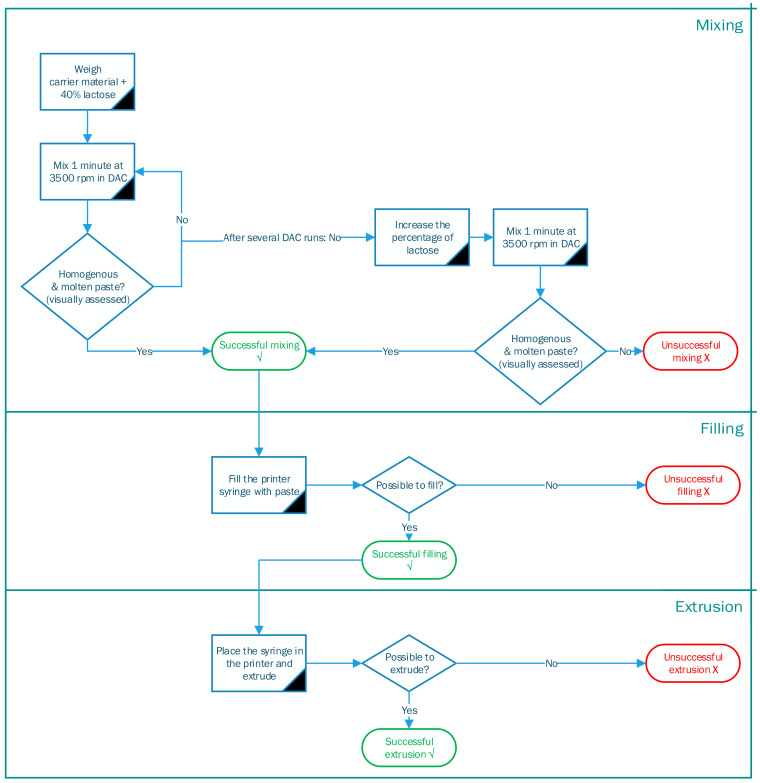
Decision tree of different processing steps for SSE 3D printing.

**Figure 2 pharmaceutics-15-00028-f002:**
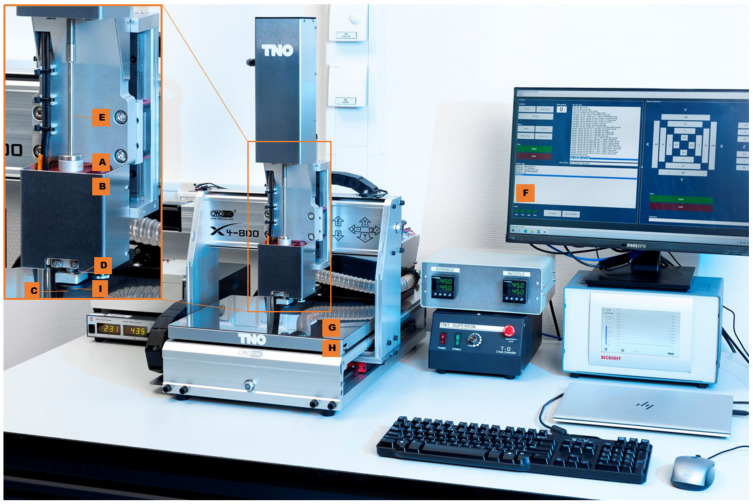
Used printer setup; syringe (**A**), the heating element of the syringe (**B**), nozzle (**C**), the heating element of the nozzle (**D**), piston (**E**), computer with Pronterface (**F**), printing platform (**G**), heating bed (**H**), and cooling (**I**).

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
