# Peer review of "The Quest for Child-Friendly Carrier Materials Used in the 3D Semi-Solid Extrusion Printing of Medicines"

_pharmaceutics, 2022, doi:10.3390/pharmaceutics15010028_

Round 1
Reviewer 1 Report
The authors detailed a systematic approach to select excipients for 3D semi-solid extrusion printing for pediatric population. Below are comments that are be addressed to improve the quality of manuscript.
1. It is understood that drug-excipient compatibility studies were not considered in this approach as it is only a preliminary selection of excipients for SSE 3D-printing. A brief statement about the importance of drug-excipient compatibility will be useful.
2. As screening studies were performed without drug, please explain if the same results can be expected with drug especially with high drug loads?
3. Introduction:
- Please revise “Low doses are needed, and, as children develop quickly, also flexibility in the dosage form is essential“ to “ Low doses are needed in comparison to adults and the dosage form needs to be flexible as children experience quick physical growth.”
- Please revise “As a consequence, a lot of manipulations occur like splitting, crushing, and dissolving tablets or administering intravenous liquids orally” to “As a consequence, a lot of dose corrections are performed through splitting, crushing, and dissolving tablets or administering intravenous liquids orally”
4. Acronyms: Certain acronyms (STEP in abstract, DAC in printability screening) need to be expanded.
Author Response
The authors detailed a systematic approach to select excipients for 3D semi-solid extrusion printing for pediatric population. Below are comments that are be addressed to improve the quality of manuscript.
- It is understood that drug-excipient compatibility studies were not considered in this approach as it is only a preliminary selection of excipients for SSE 3D-printing. A brief statement about the importance of drug-excipient compatibility will be useful.
Response: A statement on the compatibility of API and carrier material is included in the discussion (lines 332-338).
- As screening studies were performed without drug, please explain if the same results can be expected with drug especially with high drug loads?
Response: Thank you for this recommendation. We explained it in the discussion (lines 332-338).
- Introduction:
- Please revise “Low doses are needed, and, as children develop quickly, also flexibility in the dosage form is essential“ to “ Low doses are needed in comparison to adults and the dosage form needs to be flexible as children experience quick physical growth.”
- Please revise “As a consequence, a lot of manipulations occur like splitting, crushing, and dissolving tablets or administering intravenous liquids orally” to “As a consequence, a lot of dose corrections are performed through splitting, crushing, and dissolving tablets or administering intravenous liquids orally”
Response: Both lines are revised.
- Acronyms: Certain acronyms (STEP in abstract, DAC in printability screening) need to be expanded.
Response: We checked all acronyms and expanded when mentioned at first.

Reviewer 2 Report
This manuscript is a blend of a review and an experimental study. I think that neither the review, nor the experimental study are strong enough to be published. The review needs a deeper discussion about the collected data and also some parts must be made clearer. In parallel, the experimental part is very simple and needs more data regarding the accuracy of the dosage form weight (mean weight), the drug content data and better AFM images showing the liposomes in the films.
The objective of the manuscript should be also completelly revised, including the insertion of some highlights to show the importance of writing a review about this subject. In addition the inclusion and exclusion criteria were not clear for me.
This sentence should be confirmed: f 121 neither the prolonged DAC mixing nor the increased amount of lactose led to an enhanced consistency, the mixing was declared unsuccessful.
Why the carrier materials was selected based on the lower melting point of 30 oC? I could not understand and I am very curious.. rsrsrs
I would like to recommend for the authors to adjust their manuscript as a review paper. Taking into account that the manuscript needs a complete and whole revision, I would like to recommend the authors to reconsider the format of the manuscript (keeping it as a review) and avoiding experimental data in the same review.
Conclusions should be revised according to the suggested changes in the text, as commented above.
Author Response
This manuscript is a blend of a review and an experimental study. I think that neither the review, nor the experimental study are strong enough to be published. The review needs a deeper discussion about the collected data and also some parts must be made clearer. In parallel, the experimental part is very simple and needs more data regarding the accuracy of the dosage form weight (mean weight), the drug content data and better AFM images showing the liposomes in the films.
Response: This type of article combines a descriptive review and an experimental screening section. The objective of this article is to give an overview of carrier materials used in SSE 3D printing research for pediatric use and to evaluate their theoretical suitability and experimental extrudability. Drug content is not part of this article as it focus on the selection on carrier materials for pediatric use. For clarification purposes, some text about drug-excipient compatibility and role of API is added to the discussion. (Lines 332-338). We do not understand the comment on AFM and liposomes since images were not part of the research.
The objective of the manuscript should be also completelly revised, including the insertion of some highlights to show the importance of writing a review about this subject. In addition the inclusion and exclusion criteria were not clear for me.
Response: We have made some adjustments to the text that describes the objective (end of the Introduction). The selection method and inclusion criteria are described in section 2.1 - 2.3.
This sentence should be confirmed: f 121 neither the prolonged DAC mixing nor the increased amount of lactose led to an enhanced consistency, the mixing was declared unsuccessful.
Response: With this sentence the authors describe how they defined the mixing step being unsuccessful.
Why the carrier materials was selected based on the lower melting point of 30 oC? I could not understand and I am very curious.. rsrsrs
Response: We find that including rsrsrs (“Expression of laughter”) in this comment is neither appropriate nor professional. The minimal set value of 30 degrees centigrade is set as the final product should be stable at room temperature (20°C).
I would like to recommend for the authors to adjust their manuscript as a review paper. Taking into account that the manuscript needs a complete and whole revision, I would like to recommend the authors to reconsider the format of the manuscript (keeping it as a review) and avoiding experimental data in the same review.
Response: The authors preferred the current set up of the manuscript to provide a guide on the first selection of carrier materials prior to SSE printing.
Conclusions should be revised according to the suggested changes in the text, as commented above.
Response: The comments above are incorporated in the latest version of the manuscript.
Reviewer 3 Report
Thanks for the opportunity to review this paper which describe an interesting, well designed yet pragmatic and translational approach to 3D printing for children . The idea is to 1) scope all excipients potentially applicable to SSE 3DP (this part is well described) 2) short-list the ones that do not pause safety/tolerability issues for children (this needs to be improved) 3) test extrudability with their melting method (DAC - the 1st time this abbreviation is used in 2.4 (L84) should be fully described and not in 2.4.2 L107)
2.3 is way too short and do not explain the criteria used. In fact the STEP database is not explained appropriately in my opinion - as a new reader, I get very little sense of what info it provides, why and how. Maybe the results for the 3 excipients present in the STEP database should be summarised in a table?
However my understanding is that it is not because it is not in the STEP database that it can't be used as it could just be that EuPFI has not worked a these specific excipients. You could do your independent retrival of the information. Please get the full list of excipients that have been curated.
In my opinion you should consider the third one; Lactose! see https://www.ema.europa.eu/en/lactose
The use of the FDA database is also a bit obscure and should be clarified.
There is a lot of abbreviations that are not introduced properly when used first in the text eg HPMC. I would recommend to check all of them.
I was wondering in many occasion if the grade of some excipients should not be specified. eg PVP (K12), Eudragit, etc
Make sure the English is polished eg 'the use of them' is 'their use'. some verbs are in the present tense when they should be in the past tense.
Author Response
Thanks for the opportunity to review this paper which describe an interesting, well designed yet pragmatic and translational approach to 3D printing for children. The idea is to 1) scope all excipients potentially applicable to SSE 3DP (this part is well described) 2) short-list the ones that do not pause safety/tolerability issues for children (this needs to be improved)
Response: The three carrier materials which are both in described in the STEP and FDA database are fully described in the revised manuscript. See section 3.2, lines 190-195.
3) test extrudability with their melting method (DAC - the 1st time this abbreviation is used in 2.4
(L84) should be fully described and not in 2.4.2 L107)
Response: We checked all acronyms and expanded when mentioned at first.
2.3 is way too short and do not explain the criteria used. In fact the STEP database is not explained appropriately in my opinion - as a new reader, I get very little sense of what info it provides, why and how.
Response: We assume that the reviewer means section 2.2, this section is revised and the STEP database is described in more detail.
Maybe the results for the 3 excipients present in the STEP database should be summarised in a table?
Response: The three carrier materials are fully described in the revised manuscript. See section 3.2, lines 190-195.
However my understanding is that it is not because it is not in the STEP database that it can't be used as it could just be that EuPFI has not worked a these specific excipients. You could do your independent retrival of the information. Please get the full list of excipients that have been curated.
Response: The table shows a full list of carrier materials used in research on SSE 3D printing in pediatrics. We do agree that if the carrier material is not in the STEP database it cannot be used, but its toxicity should be studied carefully before using the material regularly in pediatrics. This nuance is included in the material section and the discussion of the revised manuscript. (first paragraph of the discussion, lines 266-269)
In my opinion you should consider the third one; Lactose! See https://www.ema.europa.eu/en/lactose
Response: Thank you for this suggestion.
The use of the FDA database is also a bit obscure and should be clarified.
Response: The GRAS database of the FDA gives an overview of excipients that are ‘general recognized as safe’. Thereby this database can be used
There is a lot of abbreviations that are not introduced properly when used first in the text eg HPMC. I would recommend to check all of them.
Response: We checked all acronyms and expanded when mentioned at first.
I was wondering in many occasion if the grade of some excipients should not be specified. eg PVP (K12), Eudragit, etc
Make sure the English is polished eg 'the use of them' is 'their use'. Some verbs are in the present tense when they should be in the past tense.
Response: The whole manuscript is textually revised.
Reviewer 4 Report
The manuscript describes interesting work on rationalizing the material selection steps for manufacturing of 3D-printed medicinal formulations. The work is well-described and supported by the literature. However, there is a critical issue that must be resolved, should the manuscript be considered for publication.
Minor Issue: The authors should clarify all abbreviations used, in their first appearance in text, e.g., DAC in page 2.
Critical Issue: Something that is critically missing from this work, as far as science is concerned, is the quantification aspect. Although many processes can be evaluated "by experience" visually, these evaluation steps must be (in any possible way) quantified and qualified when it comes to science and assessment of a manufacturing process, as well as when an excipient is evaluated for processability. For example, printability is an aspect that has been extensively described in the literature, providing specific parameters that can be assessed. Moreover, the precision of printed objects must be qualified, which has been also extensively described in the literature by measuring the divergence of the dimensions of printed objects, compared to the digital templates.
Author Response
The manuscript describes interesting work on rationalizing the material selection steps for manufacturing of 3D-printed medicinal formulations. The work is well-described and supported by the literature. However, there is a critical issue that must be resolved, should the manuscript be considered for publication.
Minor Issue: The authors should clarify all abbreviations used, in their first appearance in text, e.g., DAC in page 2.
Response: We checked all acronyms and expanded when mentioned at first.
Critical Issue: Something that is critically missing from this work, as far as science is concerned, is the quantification aspect. Although many processes can be evaluated "by experience" visually, these evaluation steps must be (in any possible way) quantified and qualified when it comes to science and assessment of a manufacturing process, as well as when an excipient is evaluated for processability. For example, printability is an aspect that has been extensively described in the literature, providing specific parameters that can be assessed.
Response: It is correct that the evaluation of the molten mixture was done visually by experienced operators. However, temperature is one of the parameters which can be evaluated quantitative after DAC. This is described in paragraph 2.3.2. Besides that, we did record the force of the plunger on the material while extruding, to not extend the maximal force of 400N. This is added to the revised manuscript (line 152, 255-257 and 321).
Moreover, the precision of printed objects must be qualified, which has been also extensively described in the literature by measuring the divergence of the dimensions of printed objects, compared to the digital templates.
Response: We certainly agree with this remark, but consider this practical assessment of printability parameters as the next phase in our development of the technology and formulation. The precision of the printed object is not studied, only the extrudability of the material by SSE 3D printing. This article shows a prescreening of carrier materials and not a final product and therefore the results are described qualitatively. To be more precise, the term printability is corrected in the revised text into extrudability in the parts that describe the practical experiments.
Round 2
Reviewer 2 Report
First of all, I would like to apologize for the tipos in my first report. Of course, I would never add the comment "rsrsrs" in a scientific report. It was really a mistake, which can happen to everyone (probably it was a non-intentional copy-past at that point. I fully agree with the authors that such a comment does not fit with a professional and scientific report. It had nothing to do with the content of the submitted manuscript.
However, I could not realize a significant improvement of the manuscript based on my previous comments. The revised version does not show what the authors really changed in the text, which makes our evaluation quite more difficult. I still have the opinion that the manuscript can neither be classified as a good review, nor a good experimental study (as the methodology is very simple and mainly based on the operators point-of-view). There are not any evaluation of the rheological properties, of the variation of the mean weight of the printed forms, among others. Therefore, I am still not convinced by the authors that the manuscript is suitable for publication in a high impact journal as Pharmaceutics.
Author Response
Thank you for your apologies.
We would like to take this opportunity to clarify the setup of this type of manuscript. When we were starting our work to develop a new formulation, it was an extensive search to find which carrier materials are suitable for SSE printing pediatric formulations. We started with a literature study that showed different preparation methods for semi-solid mixtures. We used that information for our further selection. Therefore we continued our quest with an experimental part. In our manuscript, we describe this journey of the literature study and experimental part.
With our manuscript, we aim to inform and guide other scientists in their pursuit of the optimal formulation for children. As our journey started in MDPI Pharmaceutics, we are confident that this is the right journal for this manuscript.
We hope that the editor sees the value of the work in the manuscript and will consider this a worthwhile contribution to Pharmaceutics. Hopefully this clarifies our point of view and our considerations in writing a manuscript in such way.
Reviewer 4 Report
The authors have successfully addressed this reviewer's comments.
Author Response
Thank you. We are looking forward to a successful publication.